# Cardiovascular Complications of Pan-Cancer Therapies: The Need for Cardio-Oncology

**DOI:** 10.3390/cancers15113055

**Published:** 2023-06-05

**Authors:** Mengjia Chen, Jianing Xue, Maoling Wang, Junyao Yang, Ting Chen

**Affiliations:** 1Department of Cardiology, The First Affiliated Hospital, College of Medicine, Zhejiang University, Hangzhou 310003, China; chenmengjia@zju.edu.cn (M.C.); 22118182@zju.edu.cn (J.X.); 3180105984@zju.edu.cn (M.W.); 2Department of Laboratory Medicine, Xinhua Hospital, Shanghai Jiao Tong University School of Medicine, Shanghai 200092, China; 3Alibaba-Zhejiang University Joint Research Center of Future Digital Healthcare, Hangzhou 310058, China

**Keywords:** cardiovascular complication, cancer therapy, cardiotoxicity, risk factor, VEGF, ICI, cardio-oncology

## Abstract

**Simple Summary:**

Worldwide, with the steady progress of pan-cancer therapy, the long-term survival of cancer patients has increased. Inevitably, along with improved life expectancy and reduced mortality, another decisive factor has also emerged—cardiovascular complications. We focus on the heterogeneity of cardiotoxicity derived from various classes of cancer therapies. Comprehensive cardiovascular risk assessment and optimal clinical monitoring prior to, during, and after treatment should be considered in candidates for initial cancer therapy. Nevertheless, the exact mechanisms of cardiovascular adverse effects remain undefined, as do specific therapeutic targets, requiring cooperation between oncologists and cardiologists, a field known as cardio-oncology. We present an updated overview of the epidemiology and mechanism underlying the major cardiac toxic effects resulting from a range of pan-cancer therapies, with emphasis on the importance of assessing relevant risk factors.

**Abstract:**

It is more likely that a long-term survivor will have both cardiovascular disease and cancer on account of the progress in cancer therapy. Cardiotoxicity is a well-recognized and highly concerning adverse effect of cancer therapies. This side effect can manifest in a proportion of cancer patients and may lead to the discontinuation of potentially life-saving anticancer treatment regimens. Consequently, this discontinuation may adversely affect the patient’s survival prognosis. There are various underlying mechanisms by which each anticancer treatment affects the cardiovascular system. Similarly, the incidence of cardiovascular events varies with different protocols for malignant tumors. In the future, comprehensive cardiovascular risk assessment and clinical monitoring should be considered for cancer treatments. Baseline cardiovascular evaluation risk should be emphasized prior to initiating clinical therapy in patients. Additionally, we highlight that there is a need for cardio-oncology to avoid or prevent cardiovascular side effects. Cardio-oncology service is based on identifying cardiotoxicity, developing strategies to reduce these toxicities, and minimizing long-term cardiotoxic effects.

## 1. Introduction

Exponential progress in the introduction of new agents has revolutionized pan-cancer treatment recently [1]. As the lifespan of cancer patients increases, globally, malignant tumors and cardiovascular diseases (CVDs) have become two of the leading causes of death [2,3]. However, potential serious cardiac and vascular adverse events related to anticancer therapy might severely affect quality of life and cause dose reduction. Accumulating clinical studies have indicated that classic chemotherapeutic agents, molecular targeted drugs, immuno-oncology agents, antiangiogenic drugs, and even radiotherapy all injure the cardiovascular system. The most typical complications of antitumor treatment targeting the cardiovascular system include thromboembolic events, hypertension, arrhythmias, myocardial ischemia or infarction, and dysfunction of the heart up to heart failure (HF) [4]. Table 1 briefly demonstrates cardiovascular complications caused by primary pan-cancer therapies.

Cancer therapeutic breakthroughs have increased the prevalence of cancer therapy-related cardiac dysfunction (CTRCD), which is generally recognized as a decline in left ventricular ejection fraction (LVEF) of ≥10 percentage points from baseline to a value < 50% [4]. Mechanistically, CTRCD is roughly divided into two distinct categories. Type I CTRCD, related to classical chemotherapeutic drugs such as anthracyclines, results in myocardial cellular destruction and irreversible damage. Conversely, Type II is distinguished by myocardial dysfunction and is usually reversible, associated with target therapy [5,6].

This review emphasizes the major aspects of cardiotoxicity in pan-cancer treatments and delineates the underlying mechanisms by which anticancer therapy potentially aggravates CVDs. Furthermore, we discuss that clinicians should identify diagnostic strategies and realize the importance of management and surveillance of cardiovascular risk factors to protect cancer patients from the toxicity. We also definitely recommend the formation of a team comprising oncologists working with cardiologists to mitigate the burden of CVDs in cancer patients. 

## 2. Cardiovascular Complications of Different Types of Cancer Therapies

### 2.1. Conventional Chemotherapy

#### 2.1.1. Mechanisms of Anthracycline-Induced Cardiotoxicity

Anthracyclines, one of the most representative and prominent examples of chemotherapies, separated from the soil microbe Streptomyces peucetius var. caesius are a class of cytostatic antibiotics that can hinder the synthesis of DNA and RNA by embedding into base pairs to form steady complexes and suppressing topoisomerase (Top) II activity, giving rise to DNA damage and inhibiting cell proliferation and metabolism [7]. Top IIβ plays a crucial role in DNA regulation by facilitating temporary single- or double-stranded breaks during vital processes such as DNA replication, transcription, recombination, and chromatin remodeling. Notably, the binding of doxorubicin to DNA and Top II isoforms leads to the formation of a ternary complex comprising Top II, doxorubicin, and DNA, ultimately resulting in the induction of double-stranded DNA breaks [8]. Interestingly, dexrazoxane is the most promising drug approved for the prevention of anthracycline-induced cardiotoxicity, and its cardioprotective effects are dependent on Top IIβ [9]. As the highly effective broad-spectrum antitumor drugs, anthracyclines are clinically widely prescribed to treat solid tumors and hematological malignancies; however, they are also well known for their cardiotoxicity [10]. The mechanisms of their cardiotoxicity tend to be highly complex, and currently, it is explicit that anthracyclines will induce damage to myocytes through the generation of free radicals [10]. For example, doxorubicin causes accumulation of free radicals in cardiomyocytes through a series of reactions, leading to lipid peroxidation of the cell membrane, damage of endoplasmic reticulum, mitochondria, and nucleic acid, and also arousing serious effluent loss of calcium in sarcoplasmic reticulum [11]. In addition, the mechanisms of cardiotoxicity may also involve the accumulation of doxorubicinol, which is a metabolite of doxorubicin in cardiomyocytes and the circulating pro-inflammatory cytokines.

#### 2.1.2. Cardiotoxicity of Anthracyclines

Cardiotoxicity caused by anthracyclines can be broadly classified into an acute or chronic one, with chronic toxicity being the most prevalent and important manifestation. Furthermore, acute cardiotoxicity can be divided into acute or subacute toxicity, and chronic cardiotoxicity can be classified into two kinds, early-onset and late-onset. However, it is crucial to emphasize that this progression represents a continuum and not just distinct phases, which provides a more comprehensive understanding of its pathophysiology. The initial injury inflicted upon myocardial cells, although often unrecognized, has been extensively documented. This injury diminishes the cardiac reserve and sets the stage for subsequent stressors that gradually lead to cardiac decompensation and varying degrees of dysfunction. Ultimately, this progressive and cumulative cardiac impairment may contribute to the delayed onset of CTRCD. 

Acute cardiotoxicity is relatively rare and typically occurs within one week after a single injection of anthracyclines or a therapy course. It may manifest as abnormal instantaneous cardiac electrical activity, myocarditis, arrhythmias, pericarditis, elevated troponin, or acute HF [12,13]. ECG (electrocardiogram) changes were observed in 20% to 30% of patients; as well, arrhythmias, including supraventricular, ventricular, and borderline tachycardia, covered 0.7% of patients, while more severe arrhythmias, such as atrial fibrillation or atrial flutter, were less common [14]. In a retrospective study of 64 patients with early-stage breast cancer, the results suggested that in all three groups, the incidence of ECG abnormalities peaked during the acute toxic reaction (within one week after the completion of chemotherapy), and the cardiac troponin T (cTnT) level within one week after chemotherapy was higher than that of various time points one year after chemotherapy (*p* < 0.05) [15]. 

Early-onset chronic progressive cardiotoxicity refers to cardiotoxicity that is detected within twelve months after the completion of chemotherapy and may continue to progress after the cessation of chemotherapy. The late-onset one occurs within decades after chemotherapy with an insidious onset. Once there comes initial acute myocardial injury, the ventricular function decreases significantly, which is usually irreversible and may manifest as arrhythmias, cardiomyopathy, and HF. Regarding adjuvant chemotherapy for breast carcinoma, the incidence of CHF after anthracycline treatment typically ranges from 0% to 1.6%, reaching up to 2.1% among patients treated with doxorubicin alongside sequential paclitaxel [16]. Hershman et al. [17] studied the cardiotoxicity of doxorubicin in elderly lymphoma patients (age, >65 years) and analyzed the association between doxorubicin and CHF by establishing Cox proportional hazards models. It was proven that the application of doxorubicin may increase the risk of developing CHF by 29%. Interestingly, among patients using doxorubicin therapy, 74% of survivors did not suffer from CHF, while the proportion of patients without doxorubicin was 79% over a span of eight years.

#### 2.1.3. Management and Follow-Up for Cardiotoxicity Induced by Anthracyclines

Risk factors associated with anthracyclines can be divided into two categories: patient-related risk factors and treatment-related risk factors [18]. Patient-related risk factors include almost any risk factor for heart damage, such as pre-existing cardiovascular disease, family history of cardiovascular disease, age (<5 or >65 years), female gender, and certain lifestyles (e.g., smoking, excessive alcohol consumption). Treatment-related risk factors included the cumulative dose of anthracyclines, which is considered the most important among all risk factors, combination with other treatments (e.g., trastuzumab, chest or mediastinal radiotherapy, and cyclophosphamide), and rapid administration of anthracyclines [19]. Cancer survivors who are at risk should undergo regular screening for traditional cardiovascular risk factors. The healthcare provider should determine the frequency of surveillance.

### 2.2. Radiation Therapy

#### 2.2.1. Mechanism of Radiation Therapy-Induced Cardiotoxicity

Radiation therapy can generate cardiotoxicity or the radiation-induced heart disease (RIHD) potentially caused by radiation disruption of endothelial barrier integrity, which induces a series of reactions such as oxidative stress, upregulation of inflammatory/pro-fibrotic factors, collagen deposition, proliferation of cardiomyocytes, myofibroblasts, and endothelial cells. Ultimately, it results in increased intima–media thickness, arterial wall lesions, and accelerated atherosclerosis [20]. In addition, radiation induces a decrease in microvascular density [21], which also leads to myocardial ischemia and oxidative stress, and, ultimately, cell death. Damaged and dead cardiomyocytes are swept up by macrophages as well as substituted by amyloid and fibrin, then calcification or scarring occurs. Some animal experiments show that radiation may affect mitochondrial function through the Nrf2 pathway, and the increase in mast cell number may play a protective role in RIHD [22]. These changes eventually lead to myocardial ischemia, HF, arrhythmia, abnormal movement of the heart wall, and in some cases, pericarditis and valvular disease [23,24].

#### 2.2.2. Cardiotoxicity of Radiotherapy in Breast Cancer

Radiotherapy is a crucial therapy for breast cancer. Darby et al. [25] suggested that radiotherapy increased the risk of primary adverse coronary events in breast cancer patients, with nearly half of the raising occurring within 10 years after treatment and lasting up to 30 years after treatment. When the mean heart dose (MHD) was 4.9 Gy (range from 0.03 to 27.72), the risk of coronary events was proportional to the MHD, and the risk climbed by 7.4% per Gy on average. A study further validated and refined the prediction model, finding that the cumulative incidence of major coronary events increased by 16.5% (95% CI, 0.6 to 35.0) per Gy of MHD over nine years after treatment [26]. Nevertheless, this linear relationship between the incidence of heart disease and MHD does not always appear to be consistent, and it is reasonable to consider that specific cardiac substructural doses better reflect the correlation between radiation and cardiotoxicity. In addition, the increase in MHD may be the result of an increase in the mean substructural dose; for example, the right ventricular dose has close-knit links with MHD [27], the incidence of cardiac adverse diseases was higher when the mean left anterior descending vessel dose exceeded 2.8 Gy [28].

#### 2.2.3. Cardiotoxicity of Radiotherapy in Hodgkin’s Lymphoma

Similar to breast cancer, patients with Hodgkin’s lymphoma (HL) exhibit cardiotoxicity after radiotherapy. Van Nimwegen et al. [29] were the first to demonstrate a linear relationship between MHD and the risk of coronary events in HL survivors, with an increase of 1 Gy in MHD related to a 7.4% growth in overall risk of coronary heart disease (95% CI, 3.3% to 14.8%). A follow-up study of pediatric Hodgkin’s disease survivors who underwent mediastinal radiotherapy revealed that symptomatic/asymptomatic heart disease occurred in 50 of 1132 patients (4.42%) after treatment, with valve defects being the most common cardiotoxic manifestation (33/1132), followed by coronary artery disease (14/1132), cardiomyopathy (14/1132), conduction disorders (10/1132), and pericardial abnormalities (8/1132) [30]. The study also suggested that the utilization of lower doses of radiation therapy could decrease the incidence of heart disease.

#### 2.2.4. Cardiotoxicity of Radiotherapy in Non-Small-Cell Lung Cancer

Radiotherapy is also an important approach for patients suffering from non-small-cell lung cancer (NSCLC). Researchers have suggested that the incidence of grade ≥ 3 cardiac events in patients with locally advanced NSCLC exceeded 10% within two years after receiving radiotherapy [31]. Atkins et al. [32] conducted a retrospective study involving 748 patients with locally advanced NSCLC that indicated that the cumulative incidence of major adverse cardiovascular events (MACE) was 5.8% and the all-cause mortality was as high as 71.3% within two follow-up years, and the risk of MACE was closely related to MHD. In another retrospective analysis containing 701 individuals diagnosed with locally advanced NSCLC, MHD exceeding 7 Gy was related to an increased one-year incidence of MACE (4.8% vs. 0%) and two-year all-cause mortality (53.2% vs. 40.0%) [33]. Remarkably, the prevalence of cardiac complications in lung cancer patients was approximately 25% to 30%, hinting that a significant proportion of patients with lung cancer may be more likely to develop MACE after radiotherapy due to pre-existing cardiovascular risk factors or heart disease history when considering RIHD [34].

#### 2.2.5. Management and Follow-Up for Cardiotoxicity Induced by Radiotherapy

Large cohort studies have shown that the incidence of symptomatic RIHD is low within the initial ten years after radiation therapy, but exhibits a notable increase thereafter. One possible recommendation entails that screening for RIHD should be conducted every five years or in the presence of symptoms, irrespective of the duration of radiotherapy. Early detection of subclinical RIHD and timely initiation of therapy may improve the long-term prognosis of cancer survivors at risk for cardiac events. Therefore, screening tests may be performed more frequently (every two or three years) after ten years from radiotherapy and annually for patients at a high risk of disease progression, such as those with coronary calcification, initial valve disease, and risk of coronary artery disease [35].

### 2.3. Targeted Therapy

#### 2.3.1. Mechanism of Trastuzumab-Induced Cardiotoxicity

Trastuzumab, a humanized anti-ERBB2 (epidermal growth factor receptor 2) monoclonal antibody (mAb), is commonly used to treat breast cancer in clinical settings by using alone or with other drugs, such as anthracycline and paclitaxel. Trastuzumab can effectively prolong the survival of patients with advanced breast cancer who are HER-2 (human epidermal growth factor receptor-2)-positive. Initially, it was believed that cardiotoxicity caused by mAbs was similar to the anthracyclines, but it was later generally classified as a Type II agent. Recent studies have suggested that, when used in combination with anthracyclines, trastuzumab contributes to and exacerbates cardiotoxicity caused by anthracyclines by affecting various cellular mechanisms related to myocardial survival and repair [36].

Currently, trastuzumab appears to have two possible mechanisms to induce cardiotoxicity. ERBB2 is vital for cardiomyocyte proliferation and function and also acts as a coenzyme of ERBB4 and NRG1, and the combination of the latter two will promote ERBB4/ERBB2 heterodimerization together with activation of ERK-MAPK and PI3K-Akt pathways, which can foster proliferation and contraction of cardiomyocytes [37]. Trastuzumab affects the growth, development, survival, and normal function of cardiomyocytes by inhibiting the assembly of key complexes involved in this process, and can also reduce the capacity of cardiomyocytes to react to stress events, leading to cardiotoxicity. Notably, anti-ERBB2 drugs are not equally cardiotoxic, such as lapatinib, which blocks epidermal growth factor receptor (EGFR), and may cause less cardiotoxic effects than trastuzumab [38]. Furthermore, trastuzumab also upregulates the ratio of pro-apoptotic proteins BCL-XS, disrupts the integrity of mitochondrial membranes, and activates apoptosis pathways [39].

#### 2.3.2. Cardiotoxicity of Trastuzumab

The cardiotoxicity of trastuzumab can manifest as HF, cardiomyopathy, and asymptomatic decreased LVEF. Trastuzumab can even lead to severe cardiac insufficiency or death. Previous clinical trials demonstrated a significantly higher incidence of cardiotoxicity including an asymptomatic reduction in ejection fraction or significant cardiac insufficiency within three years in early-stage HER-2-positive breast cancer patients receiving chemotherapy combined with trastuzumab, compared with chemotherapy alone [40]. Furthermore, the incidence of NYHA class III or IV was nearly 4% in patients receiving chemotherapy combined with trastuzumab, whereas the proportion of NYHA class III or IV heart failure in patients receiving chemotherapy alone was close to 0%. However, subsequent studies have observed a significantly higher incidence of cardiotoxicity from trastuzumab treatment compared to previous clinical trials, possibly due to different definitions of cardiotoxic disease and the fact that clinical trials tended to involve more young women.

A study of 45,537 older patients with early breast cancer found that the incidence of HF or cardiomyopathy was 26.7% in patients receiving trastuzumab alone versus 28.2% in patients receiving a combination of anthracycline and trastuzumab, and individuals treated with anthracycline alone had the lowest incidence at 15.3% [41]. The report is noteworthy for highlighting the percentage of HF or cardiomyopathy development in otherwise healthy women. Another retrospective study, which included data from 12,500 breast cancer patients, suggested that within five years of treatment, the incidence of HF or cardiomyopathy was approximately 4.3% in patients administered anthracycline alone, 12.1% in patients receiving trastuzumab alone, and 20.1% in patients receiving anthracycline combined with trastuzumab [42]. Furthermore, cardiac toxicity was also shown to be more obvious in patients receiving trastuzumab and paclitaxel than those using paclitaxel alone [43].

#### 2.3.3. Cardiotoxicity of Other HER2-Targeted Drugs

Pertuzumab, another mAb targeting HER-2, also induces cardiotoxicity. Other HER2-targeted agents, such as pertuzumab, lapatinib, trastuzumab emtansine (T-DM1), and neratinib, have been shown to have lower levels of cardiotoxicity compared to trastuzumab [44]. Pertuzumab is often used with trastuzumab in neoadjuvant therapy, adjuvant therapy, and metastatic therapy. A systematic review and meta-analysis pooling data from several studies showed no significant difference in the risk of asymptomatic/mild left ventricular systolic dysfunction between the pertuzumab and placebo groups when combined with trastuzumab, chemotherapy, or T-DM1, respectively [45]. Some studies have also indicated that pertuzumab has little additional cardiac risk for trastuzumab [44]. Approximately 0.2% of lapatinib-treated patients experienced Grade III/IV systolic dysfunction as well as asymptomatic cardiac events that occurred in 1.4%. At present, there is insufficient evidence to finalize the cardiotoxicity of neratinib and T-DM1 [46]. The incidence of cardiac adverse events did not appear to change in patients who had previously received trastuzumab, regardless of the addition of these new targeted agents [47].

#### 2.3.4. Management and Follow-Up for Cardiotoxicity Induced by Targeted Therapies

The most important risk factors for cardiotoxicity from HER2-targeted drugs are likely to be prior exposure to anthracyclines. Additional risk factors may include age, pre-existing cardiovascular risk factors, smoking, and obesity. Furthermore, trastuzumab cardiotoxicity appears to be independent of accumulated dose, which is distinct from anthracyclines [44]. Banke et al. [48] found that in nearly 10,000 patients with a median 5.4 follow-up years, trastuzumab was linked to a twofold higher risk of late HF in comparison to chemotherapy alone, despite the low absolute risk. The SAFE-HEaRt study’s long-term follow-up offers valuable and ongoing safety information regarding the use of HER2-targeted therapy in patients with compromised heart function, despite the rarity of late development of cardiac dysfunction [49].

## 3. Effects of Antiangiogenic Anticancer Drugs on the Cardiovascular System

In recent years, the anticancer angiogenesis research has developed from the early nonspecific embolization and tumor vessel occlusion to a more sophisticated and precise approach of targeted blocking of tumor vasculature. Blood vessels supply oxygen and nutrients, forming an extensive network that nourishes all of the body’s tissues. Excessive vascular development or abnormal remodeling facilitates the occurrence of several diseases, including cancer [50]. Although therapeutic methods to obstruct vascular provision have been attained in the clinic, issues with complications and limited efficacy remain. The notion put forth by Professor Folkman in 1971 that “tumor development and metastasis rely on neovascularization” has given rise to new areas of research and served as the theoretical foundation for anticancer angiogenesis medications [51]. Studies have shown that there will be a great deal of formation of new blood vessels during the growth of tumors. Neovascularization provides an approach for nutrients needed for tumor growth and excretion of tumor metabolites, and simultaneously disseminates tumor cells to other locations, creating new metastases in various areas of the body. 

### 3.1. Mechanism of Antiangiogenic Drugs-Related Cardiac Side Effects—VEGF Signaling

Vascular endothelial growth factor (VEGF) as well as its downstream signaling pathways are now recognized as the major potential target of new angiogenesis inhibitors, which are widely available in the therapy of advanced malignant tumors. Broadly speaking, tyrosine kinase inhibitors (TKIs) and mAbs are two categories of VEGF inhibitors [52]. There are four different kinds of VEGF (receptor) inhibitors available for cancer therapy: (1) bevacizumab, a humanized mAb, directs against VEGFA in the circulation [53]; (2) ramucirumab is a mAb that acts directly on cellular VEGF receptors [54]; (3) aflibercept, a soluble decoy receptor, blocks tumor angiogenesis by binding to VEGF; (4) sunitinib, sorafenib, vandetanib, pazopanib, regorafenib, axitinib, cabozantinib, fotivda and so on are small-molecule TKIs that suppress intracellular and downstream signalings [55]. 

However, given that VEGF also plays an integral role in maintaining cardiovascular homeostasis, not surprisingly, multitargeted antiangiogenic agents have unearthed a wide spectrum of cardiovascular complications, including principally hypertension, arterial thromboembolic events (ATEs), and left ventricular insufficiency, which can be fatal in some cases [56]. The fundamental causes of cardiovascular complications attribute to “on-target” effects that inhibit target tyrosine kinases expressed in the cardiovascular system on the one hand, and “off-target” effects since the agent blocks a kinase among its “unplanned” targets, which regrettably has an essential function in heart and vasculature on the other one [57]. The potential mechanisms of action of VEGF (receptor) inhibitors are schematically depicted in Figure 1. Abdel-Qadir et al. [58] conducted a comprehensive meta-analysis that quantified the cardiovascular toxicity of VEGFR inhibitions in the treatment of patients with malignancy including the increased risk of hypertension (odds ratio, OR 5.28), cardiac ischemia (OR 2.83), cardiac dysfunction (OR 1.35), and arterial thromboembolism (OR 1.52). These cardiovascular complications not only may need the adjustment, or even cessation, of life-saving anticancer treatments, with the danger of causing dose reduction, but can also greatly impact the quality of life in cancer patients [59].

### 3.2. Antiangiogenic Anticancer Drugs-Induced Hypertension

Hypertension is one of the most commonly documented cardiovascular complications of VEGF (receptor) inhibitions. Blockage of VEGF signaling decreases synthesis of nitric oxide (NO) and prostaglandin I_2_ (PGI_2_), while increasing the production of endothelin 1, which promotes endothelium-dependent vasoconstriction in venules and arterioles, ultimately affecting blood pressure. In addition, rarefaction (reduced microvessel densities), which induces impaired microcirculation and increased peripheral arterial resistance, has been proposed as a major cause of hypertension [60,61]. Treatment with VEGFR inhibitions contributes to hypertension and proteinuria by shifting the pressure natriuresis curve, resulting in reduced sodium excretion and raising podocyte permeability [62]. Hence, VEGF signaling inhibition negatively affects renal function and renovascular homeostasis that leads to hypertension [63].

The incidence of all-grade hypertension using bevacizumab ranged from 12% to 34%, with rates of grade ≥ 3 hypertension varied from 5% to 18% [64]. Bevacizumab was initially approved for metastatic colon cancer that resulted from a landmark randomized controlled trial (RCT). In this trial, 813 patients were randomly assigned to receive 5-fluorouracil, irinotecan, leucovorin, plus bevacizumab or placebo. The incidence of all-grade hypertension was 22% with bevacizumab compared with 8.3% in the placebo arm [65]. Approximately 16% of patients suffering from advanced gastric or gastroesophageal junction carcinoma with ramucirumab developed hypertension (of any grade) compared with 8% in the control arm [66]. Similar findings showed that, in cases of grade 3 hypertension or higher, the incidence was higher in the treatment group compared to the control group (8% vs. 3%) [67]. Van Cutsem et al. [68] reported that the incidence of all-grade hypertension with aflibercept was identified to be 41% compared with 10.7% in the control group, demonstrating a significant association between aflibercept and hypertension. Blood pressure is influenced by anti-VEGF medications in a dose-dependent relationship, and the degree of hypertension may serve as a useful indicator of the level of target inhibition [69]. However, the dosage limitation of a specific agent may depend on its non-VEGFR-targeted toxicity. Compared to the low-dosage group, the high-dose group showed a remarkable increase in hypertension (3% vs. 36%) among patients suffering from renal cell carcinoma who were treated with bevacizumab [70]. According to certain research, the development of VEGFR inhibition-induced hypertension may serve as a potent prognostic marker for the cancer outcome in angiogenesis inhibitor treatment [63]. As a clinical application with VEGFR inhibition treatments has expanded, it is now evident that the development of hypertension induced by anti-VEGF drugs is related to increased survival in clinical practice, although initially argued. This finding suggests that the potential mechanisms of cancer efficacy and VEGF inhibition-induced hypertension may be identical, or at least overlap. 

Although hypertension induced by angiogenesis inhibitors is common, it is generally reversible and can be effectively managed with standard medications. Proteinuria screening is recommended for all patients receiving treatment with angiogenesis inhibitors [71]. Cancer survivors should be monitored for the development of hypertension more closely than the common population, using both in-office and home blood pressure monitoring. Currently, there is not enough evidence to recommend a targeted approach for managing hypertension [72]. Preventing short-term fatal consequences of hypertension, such as myocardial infarction (MI) and cerebrovascular accident, and maximizing the clinical effect of anti-VEGF medications at therapeutic doses are the aims of antihypertensive therapy in individuals who have advanced malignant cancers with shorter survival expectancy compared with the general population [73].

### 3.3. Antiangiogenic Anticancer Drugs-Induced Ventricular Dysfunction and Heart Failure

In contrast to the cardiovascular complications caused by chemotherapy agents such as anthracyclines, the cardiotoxicity caused by antiangiogenic-targeted agents is reversible, possibly due to cellular hibernation and myocardial stunning [74]. Sunitinib, for example, lacks sufficient selectivity and blocks signaling cascades apart from VEGF [75]. Sunitinib appears to be more toxic to the heart than other angiogenesis inhibitors since it blocks more than thirty additional receptors and nonreceptor tyrosine kinases such as c-Kit, FMS-related tyrosine kinase 3 (FLT3), and platelet-derived growth factor receptors (PDGFR) alpha and beta. Inhibiting “off-target” kinases, including ribosomal S6 kinase, which triggers the endogenous apoptotic pathway, breaks energy homeostasis by 5′-AMP-activated protein kinase (AMPK) and exacerbates ATP depletion, providing an additional explanation for the increased occurrence of sunitinib-induced cardiotoxicity [76,77]. Antiangiogenic TKIs are involved in the inhibition of the Notch pathway to limit self-renewal of angiogenic cells, which may induce cardiomyocyte apoptosis and cardiac fibrosis [78]. Furthermore, the inhibition of PDGF/PDGFR can cause mesothelial cell damage, enhance vascular permeability, and affect coronary microvascular function [79]. Anti-VEGF inhibitions also act on AMPK, leading to impaired energy metabolism and mitochondrial dysfunction, which was authenticated in mice models given sunitinib and subjected to pressure load. Cardiomyocytes showed activation of opening the mitochondrial permeability transition pores, significant mitochondrial swelling, and disruption of the normal mitochondrial architecture, according to the observations of Chu and his colleagues [80].

In the case of anti-VEGF mAb, approximately 2.7% of patients with breast cancer receiving bevacizumab monotherapy experienced CHF [81]. Representing anti-VEGF TKI, the incidence of sunitinib-associated left ventricular insufficiency ranged from 10% to 13% [82]. A 1% rate of sorafenib-induced ventricular dysfunction and a 4.1% rate of sunitinib-related CHF were noticed in a pair of meta-analyses involving nearly 900 patients receiving sorafenib and 7000 patients receiving sunitinib, but the majority of these findings are derived from retrospective studies [83,84]. Few trials have incorporated prospective cardiotoxicity evaluation. Among studies using antiangiogenic agents, refractory CHF with life-threatening results has rarely been documented. 

Most patients experienced recovery of ventricular dysfunction after quitting the anti-VEGF drug. Patients receiving antiangiogenic drugs require noninvasive evaluation of their left ventricular function through biomarkers and imaging tests. Monitoring of HF symptoms is also necessary, along with addressing cardiovascular risk factors. However, long-term follow-up for asymptomatic cardiovascular toxicity is often absent, which may result in under-recognition of such cases [85].

### 3.4. Antiangiogenic Anticancer Drugs-Induced Arterial Thromboembolic Events

To explain the thromboembolic side effects caused by anti-VEGF therapy, several mechanisms have been proposed. VEGF inhibitors, by reducing the levels of PGI2 and NO, disrupt vascular endothelial integrity, expose the subendothelial basement membrane, facilitate the entry of tissue factor into the bloodstream, and ultimately activate the exogenous coagulation system [86]. Moreover, VEGF inhibitors result in vascular inflammatory responses that contribute to coronary artery disease progression and plaque rupture [87].

Bevacizumab with chemotherapy contributed to a twofold increase in ATEs (3.8% vs. 1.7%) compared with chemotherapy alone in a combined analysis of five RCTs covering 1745 candidates with three types of malignant tumors, in which VEGF inhibitors were initially identified [88]. 

The follow-up results published by the American Society of Hematology in 2014 showed that the incidence of developing overall and severe ATEs in patients using VEGF-TKIs was 19% and 14%, respectively [89]. Among these events, 10% occurred in the angiocarpy, 7% in the cerebral vessels, and 7% in the peripheral vasculature. The clinician monitors the patient’s fibrinogen, D-D dimer, and MI markers to assess the coagulation function and the risk of thrombosis in advance. When initiating antiangiogenic therapy, it is crucial to carefully evaluate both risks and benefits since most ATEs lack early warning symptoms.

## 4. Immune Checkpoint Inhibitors-Related Cardiovascular Diseases: Mechanisms, Identification and Management

Immune checkpoint inhibitors (ICIs), as emerging immunotherapeutic agents, have gained widespread acceptance and integration into clinical practice [90]. Their extensive application has demonstrated an unprecedented benefit to the anticipated survival of cancer patients [91]. ICIs are targeted in approved clinical indications including mAbs programmed cell death protein 1 (PD-1), programmed cell death protein ligand 1 (PDL-1), and cytotoxic T lymphocyte-associated protein 4 (CTLA-4). These agents block co-inhibitory molecules in T-cell activation pathways and stimulate immune responses that fight against tumors. Figure 2 illustrates the main mechanisms of ICI-associated cardiotoxicity. Unfortunately, the administration of ICIs therapy induces a broad spectrum of immune-related adverse events (irAEs) that can affect multiple organs, including the heart [92,93,94], but the exact mechanism is not fully understood. ICI-associated myocarditis and atherosclerosis are the most common and most documented in a series of irAEs in recent years.

### 4.1. ICI-Associated Myocarditis

#### 4.1.1. Epidemiology of ICI-Associated Myocarditis

ICI treatment may lead to a series of irAEs, and cardiovascular adverse events are of particular concern. Currently, major cardiovascular irAEs include myocarditis, pericarditis, cardiomyopathy, vasculitis, arrhythmias, and atherosclerosis. A retrospective study based on data from VigiBase suggested that myocarditis was found to be strongly associated with ICI treatment (ROR: 3.8 (3.08–4.62), IC_025_: 1.63), aside from which, according to previous studies, the risk of ICI-induced myocarditis is basically less than 1% [95]. Nevertheless, due to the unclear diagnosis of myocarditis and the neglect of some subclinical myocarditis in statistics, there are reasons to be convinced that the actual incidence is higher. Johnson et al. [96] showed that the incidence of myocarditis associated with nivolumab monotherapy was 0.06%, leading mortality of less than 0.1%. Conversely, the incidence of myocarditis with nivolumab and ipilimumab combined treatment was 0.27% and myocarditis mortality was 0.17%, suggesting that the incidence of ICI-related myocarditis is significantly higher in the combination therapy of ipilimumab and nivolumab compared to monotherapy. According to another retrospective study based on data from VigiBase with a bigger sample size, among 31,321 patients treated with ICI monotherapy, 122 developed myocarditis whose incidence rate is 0.39% [95]. This study also provided further evidence that the combination of the two agents raised the incidence of myocarditis and stated that, compared to anti-CTLA-4 antibody, anti-PD-1/PD-L1 antibody may be more likely to cause ICI-associated myocarditis. On average, ICI-associated myocarditis occurs approximately 30 days after the first treatment [97]. Most myocarditis symptoms are severe and can manifest as chest pain, dyspnea, myocarditis infiltrated by T cells and macrophages, early and refractory ECG changes, rhabdomyolysis, significantly elevated troponin levels, and may be accompanied by skeletal myositis and myasthenia gravis [96,98,99]. Ultimately, HF and even sudden death can occur, with a mortality rate of approximately 50% [95], the highest among irAEs [98]. Recent studies have shown that the fatality rate of ICI-related myocarditis sharply increased to 76%, and the fatality rate of combined treatment was nearly twice as high as that of anti-PD-1/PD-L1 monotherapy [97]. In addition, attention should also be given to subclinical or smoldering myocarditis after ICI treatment. The early symptoms of this type of myocarditis are atypical and can manifest as refractory nausea caused by non-gastrointestinal reasons, aside from which acute elevation of serum cardiac troponin I (cTnI) can be detected, and endocardial myocardial biopsy may show collagen fibrosis and lymphocyte inflammation [99]. In addition, the possibility that explosive myocarditis may develop from subclinical or smoldering myocarditis is not ruled out.

#### 4.1.2. Immune Mechanisms Underlying Myocarditis

PD-1 and CTLA-4 are important T-cell co-inhibitory receptors, which can inhibit T-cell activation and exert immunosuppressive functions. PD-L1 is one of the ligands of PD-1. Expression of PD-1 and CTLA-4 on tumor cells enables them to escape immune surveillance to some extent, while ICIs treatment enhances antitumor immunity and thus may treat cancer. At present, the mechanism of ICI-associated myocarditis remains to be further clarified, and T-cell immunity mediated by ICI is considered to be an important pathway of pathogenesis. In CTLA-4-deficient mice, without immunosuppression of CTLA-4, peripheral T cells were overactivated and proliferated, which mediates severe tissue damage, and the mice develop lymphoproliferative diseases with severe myocarditis as well as death within four weeks eventually [100]. The phenotype of PD-1/PD-L1-deficient mice is much more complex [98]. Due to variations in genetic backgrounds among different species, some PD-1/PD-L1-deficient mice may not exhibit a noticeable cardiac phenotype, while others may develop cardiac inflammation or dilated cardiomyopathy. Tarrio et al. [101] proposed that CD8^+^ T cells lacking PD-1 exhibited enhanced proliferation, increased secretion of granzyme B, and elevated levels of IFN-γ and IFN-γ-related chemokines, which helped CD8^+^ T cells recruit neutrophils and monocytes or macrophages inducing more target cell death and cardiac inflammation. This study also suggested that a lack of PD-1 would increase the release of IL-12 subunit p40, potentially contributing to the development of autoimmune CD4^+^ T-cell-dependent myocarditis [101]. The specific mechanism remains unclear; however, it tends to be different from the myocarditis model mediated by CD8^+^ T cells. The complementarity of different mechanisms leads to a higher frequency of myocarditis, which is consistent with the higher clinical incidence of myocarditis induced by anti-PD-1/PD-L1 therapy coupled with anti-CTLA-4 therapy. It is worth mentioning that recent studies have shown that the progression of myocarditis to HF and even death may be related to the reactions between myosin and Th17 cells [102,103].

As for which antigens trigger T-cell response in ICI-related myocarditis, there is no clear answer. Experiments have shown that tumor tissue may express high levels of muscle-specific antigens (troponin and desmin), and there may exist T cells sharing antigens between myocardium and tumor tissue [96]. Therefore, ICI treatment may cause secondary myocarditis during tumor therapy. Additionally, possible mechanisms may involve the expression of tumor antigens in cardiomyocytes or molecular mimicry [98]. 

#### 4.1.3. Management and Follow-Up in the Treatment of Cancer and Myocarditis

With the extensive application of ICIs, the mechanism of ICI-associated myocarditis will become more and more clear, and its management, treatment strategies, and follow-up will be constantly updated. At present, it is relatively clear that the risk factors of ICI-induced myocarditis are the concurrent use of ICIs with other cardiotoxic drugs [104], and the treatment of anti-PD-1/PD-L1 coupled with anti-CTLA-4 may be the strongest risk factor. Some data suggest that before ICI treatment, the ejection fraction measurement of patients could not well distinguish the group of people who are at risk of developing myocarditis [105]. The baseline assessment primarily relied on medical history, physical examination, electrocardiogram, troponin level, and echocardiography if necessary [106]. During ICI treatment, the median onset time for ICI-associated myocarditis is generally within one month, so it can be detected by continuous electrocardiogram and regular cardiac troponin level examination [107]. After ICI treatment, one approach to early detection of ICI-associated myocarditis in high-risk patients is to perform serial troponin measurements during follow-up, but there is no unified standard for follow-up time and the specific program at present [106].

### 4.2. ICI-Associated Atherosclerosis

#### 4.2.1. Epidemiology of ICI-Associated Atherosclerosis

Drobni et al. [108] retrospectively tested whether initiation of ICIs was correlated with accelerated atherosclerosis, and they observed a higher incidence of atherosclerotic CVDs in patients with cancers. The investigators found that cardiovascular events occurred in 119 patients during a two-year period, compared to 66 individuals in the two-year period prior to initiating ICI therapy, causing a 4.8-fold increase in the incidence of CVDs in this large matched cohort study [108]. Bar et al. [109] performed a single-center retrospective study and found that approximately 2.6% of 1215 oncology patients who received ICIs experienced arterial thrombotic events (ischaemic stroke up to MI) within the first six months after starting ICI treatment. Patients developing acute vascular events had a worse prognosis compared to events-free patients [109]. These checkpoints also play a crucial role in thrombosis for cancer treatment, and patients receiving ICIs have been found to have a higher incidence of venous thromboembolic events [110]. Additionally, another retrospective study covering a total of 3326 patients who were treated ICIs reported that the occurrence of MI and stroke was presented, respectively, in 6.4% and 6.8% of patients during a follow-up of 16 months on average [111]. Table 2 summarizes studies examining the association between ICIs initiation and atherosclerotic CVDs. Increasingly, clinical data that are not just limited to small cohort studies and case reports documented atherosclerosis-driven CVDs in cancer patients receiving ICIs treatment. A meta-analysis including 10,106 cancer patients receiving ICIs in 17 studies showed a 1.1% rate of the incidence of ATEs [112]. 

#### 4.2.2. Immune Mechanisms Underlying Atherosclerosis

The pathophysiology of atherosclerosis related to immune checkpoint therapies remains poorly understood. T-cell-driven inflammation is vital to the emergence and progression of atherosclerosis [113,114]. Aberrant infiltration of CD4^+^ and CD8^+^ T cells in the vascular wall is closely associated with the formation of atherosclerotic lesions, which contribute to monopoiesis and macrophage accumulation as well as necrotic core formation in the early stages of atherosclerosis [113,114,115]. In fact, CD4^+^ T-cell subsets and their effector cytokines have distinct excitatory or inhibitory effects on both tumors and atherosclerosis [115]. Treatment with anti-PD-1 and anti-CTLA-4 antibodies reduced tumor burden but increased pro-inflammatory T-cell activation, leading to atherosclerosis progression. Overexpression of CTLA-4 prevented the development of atherosclerosis by suppressing the activation of effector CD4^+^ T cells and restricting their accumulation along the arterial wall in hyperlipidemic mice [116]. 

The activation of CD8^+^ T cells is strictly regulated by immune checkpoints [113,114]. Anti-CTLA-4 antibody enhanced the motility of cytotoxic CD8^+^ T cells, while CTLA-4 prevented self-reactive pathogenic T cells from infiltrating peripheral nonlymphoid organs [117,118]. A similar phenotype was observed in PD-1^–/–^ atherosclerotic mice, which presented that abundant T cells as well as macrophages infiltrated into the vascular wall, leading to enhanced development of atherosclerotic lesions [119]. Deficiency of PD-1 signaling enhanced pro-inflammatory T-cell activation, which led to a significant increase in apoptotic cells, especially in endothelial and smooth muscle cells, which worsened atherosclerosis in Ldlr^−/−^ mice [120]. There was strong scientific plausibility to back the effects of certain ICIs on the pathophysiology of atherosclerosis inflammation [119,121,122]. Poels et al. [123] revealed that combined therapy with anti-CTLA-4 and anti-PD-1 antibodies did not influence macrophage-driven vascular or systemic inflammation in cases of melanoma and hypercholesterolemic Ldlr^−/−^ mice, whereas in a mouse model this, combination therapy increased the formation of necrotic core, aggravated cytotoxic CD8^+^ T-cell-mediated inflammation and development of atherosclerotic plaque.

#### 4.2.3. Management and Follow-Up in the Treatment of Cancer and Atherosclerosis

Candidates for ICI therapy should undergo comprehensive cardiovascular risk assessment and receive optimal clinical monitoring before, during, and after treatment to reduce the mortality and morbidity associated with atherosclerotic CVDs. The phenomenon of atherosclerosis epitomizes one of the cardiotoxic effects of ICI therapeutics, which must be placed in the context of the comprehensive care of individual cancer patients. Better clinical management and surveillance may facilitate identification of the risk factors that contribute to ICI-induced exacerbation of atherosclerosis, as well as develop effective treatment strategies. A group of researchers conducted a pilot study to investigate the cardiovascular complications associated with long-term ICI therapy (over six months) [124]. The primary endpoint, which was discontinuation of ICI due to cardiac events, was not met in any of the patients who had received single-agent ICI therapy with routine cardiology follow-up visits. However, long-term follow-up studies of patients may reveal additional insights into the enduring impacts of ICI therapy, and the effectiveness of treatment regimens can be evaluated to optimize the quality of life of the survivors.

### 4.3. Other Immuno-Oncology Agents

In chimeric antigen receptor (CAR) T-cell therapy, CAR, a synthetic receptor tailored to a specific tumor, is transduced onto autologous T cells obtained through apheresis, amplified in vitro, and then injected back into the human body to induce a targeted immune response for tumor treatment [125]. Currently, there are five CAR-T-cell therapies approved by the FDA, mainly for the treatment of leukemia, lymphoma, and myeloma. In the course of anticancer therapy, CAR-T-cell therapy has been found to have cardiovascular toxicity. 

#### 4.3.1. Mechanisms of Cardiotoxicity Induced by CAR-T Cell Therapy

The specific mechanism can be classified into three categories: “On-target, on-tumor” effect, “On-target, off-tumor” effect, and “Off-target, off-tumor” effect. Regarding the “On-target, on-tumor” effect, the infusion of chimeric antigen receptor T-cells leads to the release of pro-inflammatory cytokines and the activation of white blood cells in the body [126]. Simultaneously, tumor cell contents are released after being attacked, which also lead to the release of inflammatory cytokines [127]. Eventually, it presents as cytokine release syndrome (CRS). High-grade CRS can cause serious systemic abnormalities, including adverse cardiovascular events such as sinus tachycardia, hypotension, ventricular arrhythmias, atrial fibrillation, cardiomyopathy, and venous thromboembolism [126]. The effect of “On-target, off-tumor” is due to the fact that some normal tissues also express a certain level of tumor antigen. So the modified T cells may also attack these healthy tissues or organs, potentially resulting in life-threatening consequences, particularly if vital organs are affected [127]. However, some normal tissues may be attacked by T cells even if they do not express the same antigens as tumor cells, which is the “Off-target, off-tumor” effect. Limited studies have been conducted on the relevant mechanism, and it may be related to the cross-reaction caused by molecular mimicry of antigens.

#### 4.3.2. Cardiotoxicity and Management of CAR-T Cell Therapy

In the largest analysis of CAR-T toxicity to date, arrhythmias occurred in 2.8% of treated patients, with atrial fibrillation being the most common, followed by ventricular arrhythmias. The incidence of cardiomyopathy was 2.6%, and pericardial disease is 0.4% [128]. Before and during treatment, echocardiography, cardiac magnetic resonance imaging, biomarkers such as troponin, and electrocardiogram are important means of clinical management. Patients should be followed up at three months after treatment, and repeating these examinations after treatment is also necessary [129].

## 5. Conclusions

We provide a comprehensive analysis of cardiovascular complications induced by contemporary therapies targeting malignant tumors (Figure 3), combining a discussion of the underlying basic mechanisms with a focus on clinical management and follow-up of various therapeutic approaches. Given the growing number of indications for anticancer agents and the expansion of the application population, the number of cancer patients undergoing cancer therapy or chemoprevention with cardiovascular complications is expected to increase significantly in the future. To date, preclinical studies have not precisely identified how drugs therapy and radiation therapy affect each stage of CVDs. Further research to uncover the mechanisms that drive cancer therapies-related CVDs is essential to identify individualized interventions. At the clinical level, the correlation between cardiovascular complications and anticancer agents has not been fully established. Additional research with larger sample numbers and longer follow-up is expected to confirm this association and facilitate the screening and intervention of risk factors in high-risk populations to reduce the risk of CVDs. Given that these patients form a unique group, it is imperative to establish a comprehensive registration and follow-up system in the future to integrate the medical history, underlying disease, treatment course, cardiovascular-related adverse events, and other information of patients receiving various anticancer therapies, which is helpful to further explore and optimize the diagnosis, treatment, and prevention measures of patients with relevant cardiovascular toxicity. Clinical trials, clinical cardio-oncology programs, practice guidelines, and the establishment of specialized scientific journals, all driven by increasing awareness of cardiovascular toxicity, definitely encourage the advancement of evidence-based therapeutic strategies to mitigate potentially fatal side effects. Undoubtedly, clinicians should maximize the cardiac safety of patients during cancer therapy to improve their long-term prognosis and quality of life. With the development of cardio-oncology, synergistic effort should be required for the oncologists and cardiologists to perform evaluations pertinent to the choice of therapy. 

## Figures and Tables

**Figure 1 cancers-15-03055-f001:**
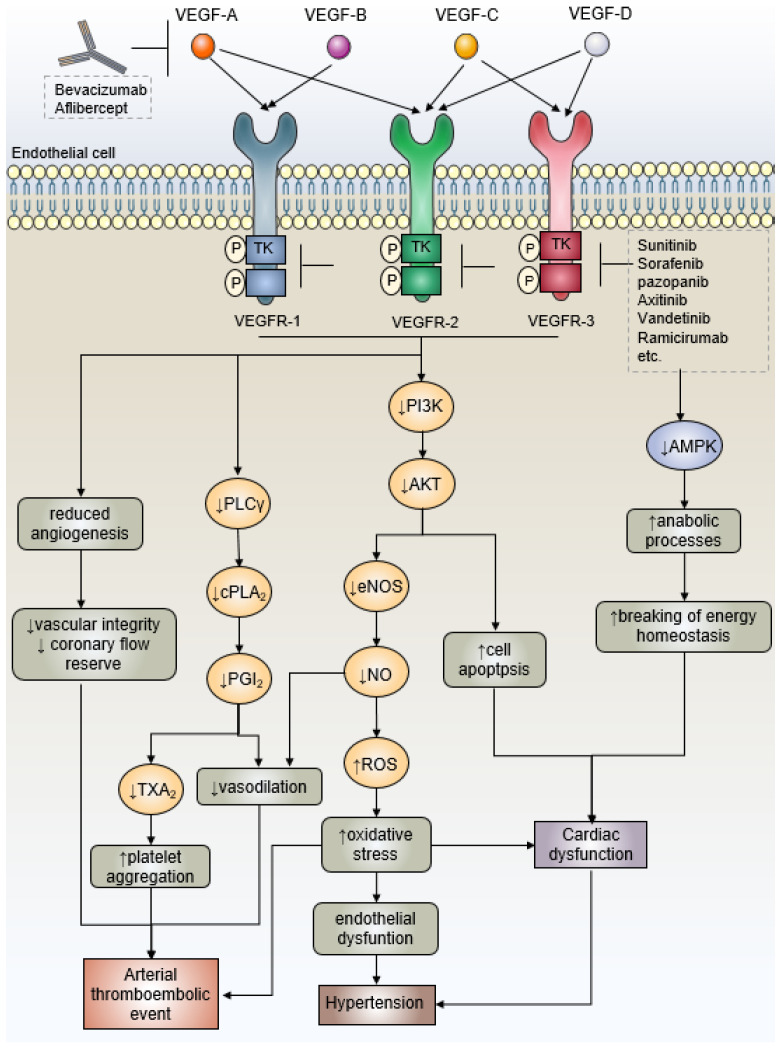
Schematic diagram of the mechanisms of VEGF (receptor) inhibitors-induced cardiovascular complications. Inhibiting the PI3K/Akt pathway causes apoptosis to be activated and ROS to accumulate, which may harm mitochondrial function. Whereas, blocking off-target kinases such as AMPK is linked to disrupting energy homeostasis, decreased cell survival, and a reduced ability to respond to cardiac stress. Each element might have an impact on how HF develops. When it comes to artery thrombosis, the lowered level of PGI2 and NO encourages platelet aggregation, and the level of TXA2 is elevated, which predisposes an individual to ATEs. Furthermore, oxidative stress may result from reduced NO production. Additionally, compromised vascular integrity could root in antiangiogenesis and lead to thrombosis. To sum up, hypertension, cardiac dysfunction, and ATEs are brought on by VEGF (receptor) suppression. AMPK, adenosine 5′-monophosphate-activated protein kinase; ATE, arterial thromboembolic event; NO, nitric oxide; PGI2, prostacyclin; ROS, reactive oxygen species; TKIs, tyrosine kinase inhibitors; TXA2, thromboxane A2; VEGF, vascular endothelial growth factor.

**Figure 2 cancers-15-03055-f002:**
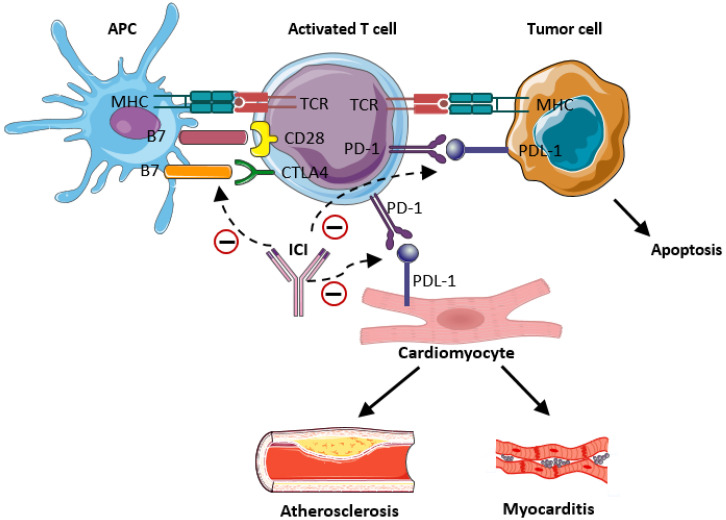
The primary mechanisms of ICIs-associated cardiotoxicity. Surface MHCs and co-inhibitory molecules including B7 or PD-1 ligands on tumor cells as well as APC (e.g., dendritic cells) attach to activated T cells through the TCR, PD-1, and CTLA-4 receptors. Apoptosis of tumor cells induced by ICI is mediated by the activation of T cells. The PD-L1 ligands on cardiomyocytes are hypothesized to be responsible for downregulating this pathway in the myocardium. Myocarditis and atherosclerosis are the primary outcomes of blocking this regulatory pathway. APC, antigen-presenting cell; CTLA-4, cytotoxic T lymphocyte-associated protein-4; MHC, major histocompatibility complex; PD-1, programmed cell death 1; PD-L1, programmed death ligand 1; TCR, T-cell receptor. Figure 2 was modified from Servier Medical Art (http://smart.servier.com/ (accessed on 30 March 2023)), licensed under a Creative Common Attribution 3.0 Generic License (https://creativecommons.org/licenses/by/3.0/ (accessed on 30 March 2023)).

**Figure 3 cancers-15-03055-f003:**
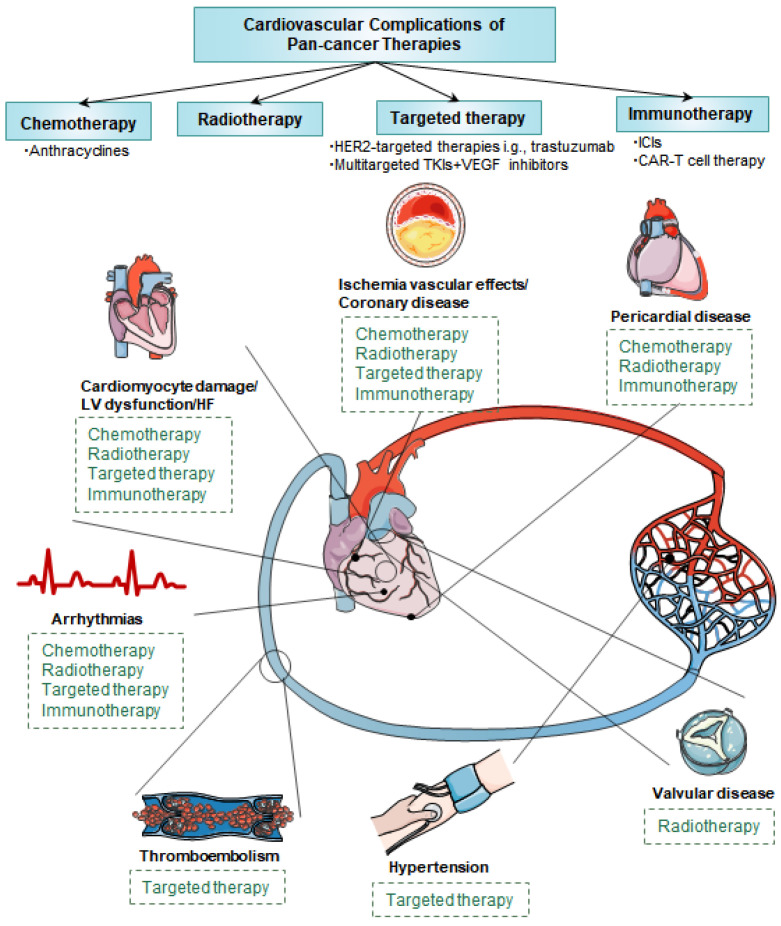
An overview of cardiovascular complications caused by various cancer therapies. There exists a plethora of acknowledged cardiovascular side effects associated with cancer treatments. Pan-cancer therapies encompass a range of treatments, including classic chemotherapeutic agents, targeted drugs, immuno-oncology agents, and radiotherapy, each impacting various aspects of the cardiovascular system through distinct mechanisms. CAR, chimeric antigen receptor; Her-2, human epidermal growth factor receptor-2; HF, heart failure; ICI, immune checkpoint inhibitor; LV, left ventricular; MI, myocardial infarction; TKI, tyrosine kinase inhibitor; VEGF, epithelial growth factor receptor.

**Table 1 cancers-15-03055-t001:** Cardiovascular complications induced by main pan-cancer therapies.

Anticancer Treatment		Classical Drugs	Main Cardiovascular Complications	Mechanism
Chemotherapeutics		Anthracyclines	Arrhythmias, ECG changes, HF, pericarditis, myocarditis, acute MI	Free-radical-mediated myocyte damage, lipid peroxidation of cell membrane, the accumulation of drug metabolites, the damage of mitochondria and DNA, sarcoplasmic reticulum stress, the release of circulating pro-inflammatory cytokines, the activity on drug transporters, Top I and II inhibition, effluent loss of calcium in sarcoplasmic reticulum.
		5-Fluorouracil	Angina and MI, hypotension, arrhythmias, ECG changes	↑Coronary vasospasm, ↑Oxidative stress.
		Cisplatin	Angina and MI, arrhythmias, chest pain, hypotension	Cytotoxicity in endothelial cells, increasing of free radicals, mitochondrial damage, calcium ion disorder.
Radiotherapy		/	Lesions of vascular segments present (e.g., accelerated coronary artery disease, stenotic aortic lesion), MI, HF, arrhythmias, pericarditis, valvular disease	Radiation disruption of endothelial barrier integrity, oxidative stress, upregulation of inflammatory/pro-fibrotic factors, decline in microvascular density, affecting mitochondrial function through Nrf2 pathway.
Targeted therapy	Anti-ERBB2 monoclonal antibody	Trastuzumab	LV dysfunction/HF, arrhythmias, hypertension, thromboembolism	Restraint of ERK-MAPK and PI3K-Akt pathways, oxidative stress, upregulation of the ratio of pro-apoptotic proteins BCL-XS disrupting mitochondrial membrane integrity and activating apoptosis pathways.
		Pertuzumab	LV dysfunction/HF	Restraint of ERK-MAPK pathway, inhibition on AMPK, disorder of sarcomeres/myofibers.
	Anti-VEGF/VEGFR inhibition	Sunitinib, sorafenib	Hypertension, LV dysfunction, thromboembolism, arrhythmias (e.g., QT prolongation), MI, bleeding	Mitochondrial damage, endothelial dysfunction, retention of sodium and water, capillary rarefaction, oxidative stress, coronary vasospasm, hypercoagulability.
Immuno-oncology agents		CTLA-4 inhibitor, PD-1 and PDL-1 inhibitor	Myocarditis, vasculitis, pericarditis, arrhythmias (e.g., supraventricular and ventricular tachycardia, heart block), and atherosclerosis	↑T lymphocyte proliferation targeting homologous antigens shared by both the tumor and myocardium, macrophage infiltration→infiltrating into cardiomyocytes by activated T lymphocytes→the inhibition of innate immune protective mechanisms in the heart→inflammation and injury. ↑Atherosclerotic inflammatory activity and progression of atherosclerotic plaque volume.
		Yescarta, Kymriah, Tecartus	LV systolic dysfunction, ECG changes, supraventricular arrhythmias (e.g., sinus tachycardia, atrial fibrillation), pericardial effusion, cardiogenic or vasodilatory shock, refractory hypotension, cardiomyopathy, and cardiac arrest	“On-target, on-tumor” effect (transferred T cells are activated and tumor cell contents are released after being attacked→multiple cytokines are released→cytokine release syndrome and cytotoxic effects on cardiomyocytes). “On-target, off-tumor” effect (T cells attack normal tissue that shares some similarities with tumor). “Off-target, off-tumor” effect (T cells attack normal tissue with certain antigens which are cross-creative with the tumor antigen, and it is related to molecular mimicry of antigens).

Abbreviations: AMPK, adenosine 5’-monophosphate-activated protein kinase; CTLA-4, cytotoxic T lymphocyte-associated protein 4; ECG, electrocardiogram; ERBB, epidermal growth factor receptor; HF, heart failure; LV, left ventricular; MI, myocardial infarction; PD-1, programmed cell death protein 1; PDL-1,programmed cell death protein ligand 1; Top, topoisomerase; VEGF, epithelial growth factor receptor.

**Table 2 cancers-15-03055-t002:** Studies examining the association between ICIs initiation and atherosclerotic events.

First Author (References)	Patients	Study Types	Follow-Up	Main Findings	Conclusions
Drobni et al. [104]	2842 (patients)/2842 (controls, a 1:1 ratio matching based on respect age, CVDs history and cancer type)	Retrospective single-center matched cohort study and case-crossover analysis	a median of 5 cycles/2 years	In the matched cohort study, the incidence of CVDs was three times higher post-ICI when compared to controls (HR 3.3, 95% CI 2.0–5.5, *p* < 0.001). In the case-crossover analysis, CVDs rose from 1.37 to 6.55 per 100 person-years within 2 years (adjusted HR 4.8, 95% CI 3.5–6.5, *p* < 0.001).	Following the start of ICIs, CV events increased, maybe caused by accelerated atherosclerosis development. Optimal CV risk factors should be considered during the whole process of treatment.
Bar et al. [105]	1215	Retrospective single-center cohort study	6 months	Approximately 2.6% (95% (CI): 1.8–3.6) of patients who did receive ICIs developed an arterial thrombotic accident (MI or ischaemic stroke). Survival of patients with acute vascular events had a worse prognosis than that of those without events.	ICIs initiation augmented the risk of acute vascular events, suggesting that caution should be exercised for ICI-related risk factors.
Gong et al. [106]	2854	Retrospective single-center cohort study	2 years	The rate of VTE increased by more than 4-fold after ICI treatment set on (HR 4.98, 95% CI 3.65–8.59, *p* < 0.001)	Patients who have taken an ICI have a high rate of VTE.
Oren et al. [107]	3326	Retrospective single-center cohort study	a mean follow-up of 16 months	The incidence of MI and stroke was presented, respectively, in 213 (6.4%) and 227 (6.8%) patients.	In cancer patients receiving ICIs, CV factors are correlated with clinical outcomes and may be utilized to estimate mortality.

Abbreviations: ICI, immune checkpoint inhibitor; CV, cardiovascular; CVD, cardiovascular disease; HR, hazard ratio; CI, confidence interval; MI, myocardial infarction; VTE, venous thromboembolism.

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
