# Peer review of "Cardiovascular Complications of Pan-Cancer Therapies: The Need for Cardio-Oncology"

_cancers, 2023, doi:10.3390/cancers15113055_

Round 1

Reviewer 1 Report

The review is well organized and written. Introduction is on point. Description of CV complications in chemio and radio therapy is on point. Conclusion is sufficient. I would only add a general consideration on clinical management and follow up. 

English language requires minor fixes.

Reviewer 2 Report

This manuscript deals with the main cardiovascular complications (CVC) following cancer treatment. The authors deal sequentially with the effects of chemotherapy, radiotherapy treatment, antiangiogenic drugs and immune checkpoint therapy and their CVC. For each of these, clinical data of incidence, molecular mechanisms of development and patient management are treated.

The review is well written as a whole even if it reports data and arguments already abundantly covered in other manuscripts, bringing no novelties in the literature in terms of future ideas. 

However, there are a few points to review below:

The first part on chemotherapy is too long and wordy and needs to be shortened.

Table 1 on Immuno-oncology, is not very clear and the concepts of 'on-target on-tumor' or 'off-target off-tumor' are not well described in the manuscript and need a more detailed description.

The chapters are badly numbered and this generates a lot of confusion during reading. This numbering absolutely needs to be revised.

A final image or cartoon that summarizes all the mechanisms described for cardiovascular complications would help in understanding the text.

There are also minor inaccuracies in the text:

some references are poorly localized in the text.

the term etc appears several times, this is not suitable for a scientific text, the lists and the complications identified must be fully reported.

Line 643 Raffaella et al., be careful raffaella is the name and not the surname of the author.

Reviewing all these little things is essential for the acceptance of the manuscript.

Minor revision needed.

Reviewer 3 Report

Is the comment at line 23 of the abstract an overstatement? You note “One of the most dramatic side effects of cancer therapies is cardiotoxicity…” Consider re-statement.

At line 51 you note: “10% absolute drop” Consider using the term a drop of 10 percentage points for clarity.

Line 55 reference 8 incorrect for type I and II. The correct reference is JCO 2005 May 1;23(13):2900-2. doi: 10.1200/JCO.2005.05.827. 

Line 95: Incorrect reference 15. The cited reference has 160 cases, you note 682. Also, reference 15 is over 30 years old, and addresses interventions no longer considered therapeutic.

Section 1.3. Risk factors: The brief description at lines 132-137 does not reflect the modern view, that anything that puts a strain on the heart or has damaged the heart is a risk factor. Consider material in any of the recent books on cancer and the heart.

Lines 140-148 might be revised to reflect that trastuzumab was initially thought to have cardiotoxicity similar to the anthracyclines until it was later shown to be a Type II agent.  To the knowledge of this reviewer, no EM changes have been found in humans. It is now thought to impair myocardial recovery after anthracycline exposure, making it troublesome when used concurrently with or soon after anthracyclines.

Line 373 has a typo: “…more toxic of heart among than angiogenesis inhibitors since it…”

In section 2.4 you address management. Consider leaving specific recommendations to organizations charged with the task of recommendations regarding dosages and monitoring.

There are several instances where the English is awkward. Careful editing will find these. As examples, this reviewer notes:

Doxorubicin was administered to treat the whole patients. (Line 97)

The study also suggested that the utilization of lower doses of radiation therapy could descend the incidence of heart disease. (Line 248)

Round 2

Reviewer 2 Report

The authors fully responded to my comments.

Author Response

Dear Reviewer,

Thank the reviewer for his/her constructive feedback on our manuscript. We appreciate his/her acknowledgment of our clear presentation of clinical data, molecular mechanisms, and patient management strategies for the cardiovascular complications associated with cancer treatments. In this paper, we present an updated overview of the epidemiology and mechanism underlying the major cardiac toxic effects foucs on a range of pan-cancer therapies. We appreciate the reviewer's valuable input and suggestions for improvement.

We would like to once again thank the reviewer for careful reading of our paper, and for the helpful comments and suggestions.

Sincerely,

Ting Chen, PhD

Professor of Cardiovascular Medicine

Reviewer 3 Report

Your comment regarding response 3 to the initial review of this reviewer is enigmatic, in that the classification of Type I and Type II, notes that agents that cause direct cell injury and death (Type I) are different from those that cause dysfunction which usually is secondary in nature. This fundamental difference is now universally recognized and is, by no means archaic. You mention “Type II” when you discuss trastuzumab, and in that instance the reference you removed (your prior reference 8) would be correct there. My suggestion is that you keep this distinction in, as it add to the understanding of important differences between direct and indirect toxicity that will help your readers understand this fundamental difference. 

You might consider strengthening the topoisomerase story, as the single most effective strategy for prevention of anthracycline toxicity is dexrazoxane that is mechanistically connected.

Your description of anthracycline looks at “acute” and “chronic,” but you might strengthen the concept that this is a continuum; the manifestation of the actual injury that injures or destroys the myocytes often is not recognized but has been well documented. This damage leaves the heart with less reserves and sequential stress over time leads to decompensation and ultimately degrees of dysfunction that are seen on imaging and that ultimately may cause the delayed presentation of heart failure. You may want to bring that into more modern thinking. 

In section 2.2.5 you note guidelines. While they may be prudent, there is yet no data that clearly demonstrates a clinical benefit to strict adherence to these guidelines known to this reviewer. You may want to comment.

Round 3

Reviewer 3 Report

You have addressed the noted issues.